Anime popularity prediction before huge investments: a multimodal approach using deep learning

Armenta-Segura Jesús
http://orcid.org/0000-0003-3901-3522 Sidorov Grigori sidorov@cic.ipn.mx
Centro de Investigación en Computación, Instituto Politécnico Nacional , Mexico City, Mexico City , Mexico
Bolshoy Alexander
Electronic publication date: 2025 Jun 2
Publication date: 2025
Volume: 11
Electronic Location ID: e2715
Received 2024 Jul 2; Accepted 2025 Jan 27
Copyright: © 2025 Armenta-Segura and Sidorov
Copyright year: 2025
Copyright holder: Armenta-Segura and Sidorov
License: This is an open access article distributed under the terms of the Creative Commons Attribution License, which permits unrestricted use, distribution, reproduction and adaptation in any medium and for any purpose provided that it is properly attributed. For attribution, the original author(s), title, publication source (PeerJ Computer Science) and either DOI or URL of the article must be cited.
License URL: https://creativecommons.org/licenses/by/4.0/

Keywords: Anime, Entertainment, Regression, Multimodal, Computer vision, Natural language processing, Popularity prediction

Funding: Mexican Government through the National Scholarship Program of CONAHCYT A1-S-47854 Secretaríıa de Investigación y Posgrado of the Instituto Politécnico Nacional, México 20241816, 20241819 and 20240951 The work was done with support from the Mexican Government through the national scholarship program of CONAHCYT, Mexico, granted to Jesús Armenta-Segura, grant A1-S-47854 of CONAHCYT, Mexico, grants 20241816, 20241819, and 20240951 of the Secretaríıa de Investigación y Posgrado of the Instituto Politécnico Nacional, México. The funders had no role in study design, data collection and analysis, decision to publish, or preparation of the manuscript.

==============================
In the Japanese anime industry, predicting whether an upcoming product will be popular is crucial. This article introduces one of the most comprehensive free datasets for predicting anime popularity using only features accessible before huge investments, relying solely on freely available internet data and adhering to rigorous standards based on real-life experiences. To explore this dataset and its potential, a deep neural network architecture incorporating GPT-2 and ResNet-50 is proposed. The model achieved a best mean squared error (MSE) of 0.012, significantly surpassing a benchmark with traditional methods of 0.415, and a best R-square (R2) score of 0.142, outperforming the benchmark of −37.591. The aim of this study is to explore the scope and impact of features available before huge investments in relation to anime popularity. For that reason, and complementing the MSE and R2 metrics, Pearson and Spearman correlation coefficients are used. The best results, with Pearson at 0.382 and Spearman at 0.362, along with a well-fitted learning curves, suggests that while these features are relevant, they are not decisive for determining anime popularity and they likely interacts with additional features accessible after further investments. This is one of the first multimodal approaches to address this kind of tasks, aiming to support an entertainment industry by helping to avoid financial failures and guide successful production strategies.

Introduction

Anime is defined as animation made in Japan. On its way to developing an identity, it has acquired several stereotypes and clichés that have made it quirky and recognizable, establishing a strong sense of identity. This cultural phenomenon has ultimately led to the formation of one of the most profitable entertainment industries in the world: the anime industry.

As in any other industry, one of the most crucial aspects on anime is the release of successful and profitable products. In order to achieve this, several techniques has been refined and perfectioned such as cultivating a fan base, where companies foments the popularity of the product among a certain demography through marketing campaigns. This practice ensures a basement of devoted customers, significantly reducing the sales risk. Since anime is a cultural phenomenon, each aspect of its success must be linked to its reception among a demography (corresponding to a cultural block) and hence the measuring and prediction of popularity of an incoming anime arises as one of the most important and relevant tasks with respect of the industry. The successfully prediction of the popularity of a future project may prevent a financial catastrophe, but also can aid on the creation of relevant, profitable and successful franchises.

When addressing this task, it comes to light that there are currently no straightforward methods to predict the popularity of an incoming anime. For instance, the Toei Animation’s 1994 film Dragon Ball Z: Bio-Broly lost more than ¥9 billion (Wikipedia, 2024; Fandom, Inc, 2025), despite being part of the globally acclaimed Dragon Ball Z franchise and featuring Broly, one of its most popular antagonists. Conversely, MAPPA Studios’ 2018 anime Kimetsu No Yaiba, based on a relatively unknown manga not among the top 50 best-selling mangas in the Oricon 2018 ranking (Oricon News, 2018), achieved unprecedented success with its 2020 film Kimetsu no Yaiba: Mugen Train, setting a worldwide Guinness Record at the box office (Pilastro, 2023), even amidst the challenges posed by the COVID-19 pandemic. These two examples show that the current understanding about the phenomenon is not yet enough to design deterministic and straight approaches for efficient and accurate popularity predictors, and hence encourages the use of heuristic approaches, such as machine learning (ML) and deep learning (DL) methods.

Another important constraint to consider is the limitation of accessible features for determining popularity. Recall that the key issue for the anime industry is avoiding financial and cultural failures. Therefore, the first challenge to address is to evaluate whether it is possible to predict popularity using only features available before making huge monetary investments. This consideration drastically narrows the possibilities, since animation has an expensive nature: even a small project such as the pilot of the indie animation HEATHENS (from Australia, but leveraging the style, cliches and quirks of anime) is projected to cost $39,460 USD (or $60,000 AU) according to their crowdfunding webpage (Kickstarter, 2024). Consequently, accessible features for this first approach cannot involve critical aspects such as a written script, animation frames or soundtrack. Instead, it has to focus solely in a brief description of the plot, along with main character sketches and descriptions (Watanabe, 2020), as it happens in other entertainment industries like Hollywood, where financial decisions must be taken based on a concise four-line description of the future movie plot (Field, 2005). This real-life investment experiences set the guidelines for the dataset construction, but also conducts to the formulation of the main research question of this article:

RQ: What are the scopes and impact of the available features before making huge investments, in relation of the popularity of an anime?

The objective of this work, hence, is to measure how influential such features can be with respect of popularity. For that reason, a dataset is proposed along with a deep neural network who embeds the features through transformers and residual networks, and then process them with a multilayer perceptron (MLP) in order to match its dimentionality with the popularity measure of the dataset, making them comparable.

With respect of the dataset, it was gathered from an internet social network named MyAnimeList (MAL) (MyAnimeList Co., Ltd, 2025b), aiming at obtaining a meaningful measure for popularity across real users. In this social network it is possible to find user ratings for animes, but also free equivalents of the limited features discussed previously: plot synopses as free substitutes for the four-line plot summaries and in-anime snapshots with fan-made descriptions as free substitutes for the main character sketches. Moreover, MAL it is the best social network of internet with respect of the objectives of this work since is the most visited anime webpage of the internet, according to https://www.similarweb.com/website/myanimelist.net. Also, when compared with their two principal competitors AnimeNewsNetwork’s encyclopedia (AENC) (Anime News Network LLC, 2025) and AniList (AL) (Python Imaging Library (Pillow’s Fork), 2022), MAL’s catalog arises as the cleanest and soundest. In the case of AENC, it encompass more than 28,000 animes but has low user interactions, while AL only includes around 15,000 animes but, although includes detailed statistics, it still underperforms MAL, who has more than 25,000 animes with detailed information about user ratings for a considerable subset of them.

With respect of the deep neural network, the embeddings are fine-tuned using the mean squared error (MSE) loss function, which compares the MAL score (measure of popularity, and the observed variables) with the output of the final MLP (it must be 1-dimensional). If the learning curves shows a proper fit, the correlation between the predictions and observed variables can provide a reasonable measure of the scope and impact of the features with respect to the popularity of an anime. High correlations would suggest a total impact and full scope, meaning these features have a capital importance with respect of popularity. Moderate correlations would indicate partial impact and moderate scope, implying the existence of a synergy with others features that may become accessible after further investment. Low correlations, along with a good fit, would suggest no scope and an absence of impact for determining popularity, meaning that this phenomenon depends entirely on features that require additional investment to access.

In order to select suitable components for the deep neural network and ensure the soundness of this study, it is important to review related and previous work. Although there is no publicly available research specifically on anime popularity prediction, three types of relevant related work can be identified: AI applications in anime, recommender systems and popularity prediction tasks in other entertainment industries.

In the first category, notable works include AI applications in sentiment analysis for anime. For instance, early attempts can be found in AlSulaim & Qamar (2021), when the authors proposed a method for sentiment polarity analysis over anime reviews from a Kaggle dataset. Their data was highly unbalanced, with 93% of positive samples vs. only 7% of negative reviews, so they applied data augmentation to achieve balance, although their explanation of the process was vague and lacked of examples, making it difficult to determine whether their model classified according to sentiment polarity (positive or negative) or according to augmented vs. non-augmented reviews. Other and more elaborated and sound approaches can be found in Théodose & Burie (2023), where the authors introduced a dataset for facial emotion recognition, while in Sharma & Chy (2021), the authors tackled the task of anime genre prediction using web snippets as the main feature. This latter study leverages deep learning techniques to address a complex task, achieving promising results from a very small feature set, which suggests that a similar approach could be beneficial here.

In the second category, recommender systems consists on methods that suggests the best possible products to potential users or customers (Ricci, Rokach & Shapira, 2022). When such systems are profile-based, they become deeply related to the popularity prediction task since they can determine the probability of a product being a suitable option for a consumer given their demographics or, in other words, how successful the product might be for that particular user/demographic. However, although tackling the popularity prediction task through (adapted) recommender systems may be tempting, they focus on an large set of demographics which lead to a significant sparsity on their datasets (Yu et al., 2024). Moreover, these datasets must include features only accessible once the product is released, such as Producers, Duration, or Casting. Such features have shown a considerable correlation with success (Wang et al., 2019; Airen & Agrawal, 2023), but are only available in the context of a recommender system for a streaming service and hence its consideration goes beyond the scope of this work.

Finally, for the third type, it is possible to find several proposals on other related entertainment industries, such as movies (Kim, Cheong & Lee, 2019) or books (Maharjan et al., 2018; Sharma, Chakraborti & Jha, 2019; Wang, Zhang & Smetannikov, 2020) and even in the anime industry itself (Armenta-Segura & Sidorov, 2023). All these works have in common the employment of relatively small models with a low-to-medium complexity, but also to consider the task as a classification problem, with artificially crafted dependent variables in order to simplify the problem. From these experiences it is possible to distill the need of large and complex models as showed in Armenta-Segura & Sidorov (2023), where the authors employed traditional classifiers over anime plot summaries and demonstrated the huge underlying complexity hidden on this kind of data, or as implicitly showed in Kim, Cheong & Lee (2019), where the authors employed a two-branched neural network with ELMO and BiLSTM or CNN to predict binary movie success through plot summaries, but then enhanced their results by leveraging BERT-based models in Lee, Kim & Cheong (2020). As a conclusion, while recommender systems have the need of larger and less sparse dataset, popularity prediction systems have the need of larger and deeper models.

In alignment with this distilled knowledge, any relevant approach must focuses on deep neural networks with complex components. By recalling that the dataset includes text and image features, the model must to be also multimodal. With respect of text inputs, the current state of the art are undoubtedly transformer-based models (Vaswani et al., 2017). This work opted for leveraging GPT-2 (Radford et al., 2019) for text analysis, 124M-parameters version. With respect of images, there are several state of the art encoders such as residual networks (He et al., 2015). This work employed the Microsoft implementation of ResNET-50, available at Huggingface, for image processing. Furthermore, this work also presents a naive benchmark for the dataset utilizing traditional vectorizations, such as TF-IDF for texts and PILToTensor for images, aiming to reproduce the work from Armenta-Segura & Sidorov (2023) and its extended version (Armenta-Segura, 2023).

In the remaining sections of the article, all described procedures are examined in detail. The obtention of the corpus is explained, alongside the experimental setup of a three-input deep neural network used to make the experiments. As previously hinted, the accuracy of the models are evaluated through MSE. Additionally, the correlations for answering the research question are measured through the Pearson and Spearman correlation coefficients, who also brings a most flexible evaluation of performance by checking the likelyhood of a linear mapping between the predictions and the real scores, either in a rank and a rankless fashion. Following this line, R squared (R2) score is also provided for evaluate the error variances, although their relevance must be understood with respect of the fit of the learning curves. The best results were obtained by considering all inputs, yielding a MSE of 0.012, a Spearman correlation coefficient of 0.362, a Pearson correlation coefficient of 0.382 and a R2 Score of 0.142 which, along with the good fit of the learning curves, permits to conclude the existence of partial impact and moderate scope of available features before huge investments, and hence the existence of a synergy between these features and others that may become accessible after further investment. The detailed study of such synergy is left as further work.

Methodology

This section presents the detailed procedure for obtaining the corpus, along with its relevant statistics. It also presents an explanation and justification of the popularity metric used (the MAL’s weighted average score), including its scope and limitations. Additionally, it presents the technical specifications of the deep neural networks and traditional benchmark models. Portions of this text were previously published as part of a preprint (Armenta-Segura & Sidorov, 2024).

The anime corpus

For each anime in the MAL database, a python script scrapped its title, plot summary, weighted average score and all of its main character names, descriptions and portraits. All samples without this information were discarded. The scrapping process started at December 28, 2023, at 0:03 UTC, and finished at January 3, 2024, at 14:31 UTC. The script was implemented using the BeautifulSoup4 python library (Richardson, 2007). The final result was 11,873 animes with 21,329 main characters.

Once obtained the data, a clean process was performed. First, all characters with no useful descriptions, such as the text Nodescriptionavailable, were removed, as well as characters with no portrait. Then, all animes with no score, synopsis, title or associated main characters were also removed, including samples with plot summaries shorter than 20 words. This process reduced the scraped data to 7,784 animes and 14,682 characters. The characteristics of the final dataset and its statistics are presented in Table 1 and Fig. 1 with respect of their synopsis and in Table 2 and Fig. 1 with respect of their main characters.

Table 1 Statistics for the corpus of the synopsis.

Wordcount refers to synopsis. Notice that the column Samples hints a normal distribution.

Score	Samples	Max. Words	Min. Words	Avg. Words	
1–2	2	68	55	61.50	
2–3	6	228	25	112.50	
3–4	11	166	24	80.72	
4–5	80	244	25	78.88	
5–6	925	329	24	83.42	
6–7	3,131	397	24	97.35	
7–8	3,021	581	24	121.30	
8–9	596	340	29	149.68	
9–10	12	189	133	157.25	
Total	7,784	581	24	104.73	

Figure 1 Mean synopsis wordcount (left) and mean main character frequency (right) across the dataset.

The X-axis represents the floor MyAnimeList score. Y-axis is the mean words on synopsis per score and mean amount of main characters per score.

Table 2 Statistics for the characters.

Total characters	Max. Words	Min. Words	Avg. Words	
14,682	3,551	4	121.34	

The MAL weighted average score as golden label

The most notable MAL statistic for popularity measuring is the weighted average score, calculated in terms of all user ratings of each anime. This score (MyAnimeList Co., Ltd, 2025a) works straightforward: a user of the database can ranks an anime in a 0-out-of- 10 scale, based on their particular opinion about it. To avoid bots, the system requires the user to watch at least a fifth part of the anime before score it, which can be done by manually marking the episodes as already seen. From this, it is possible to propose a naive measure for popularity as the mean of all rankings:

(1) S=Sumofallusersscoresoftheanimev.

where v is the total number of voters. However, this measure can be biased since does not consider the statistical relevance of the population who scored it. An example of a biased score can be a hypothetical incoming anime who receive all of its first ratings from its producers, which might be interested to give generous scores, regardless the true scope of their product. To tackle this bias, MAL weighted S in terms of how many people has watched and scored the anime. They defined a statistical bound m=50 and they defined the weight of S as:

(2) s=(vv+m).

Hence, if more people scores the anime, s tends to 1 and the relevance of S increases. When v=1, the weight reaches its minimum nonzero value 1/(m+1), who is also the average statistical importance of a single rating.

MAL also weights the general importance of their whole community, by calculating twice per day the follow default score, based on all rankings across the database.

(3) C=SumofallvalidscoresinthedatabaseTotalamountofvalidscoresinthedatabase.

The value of C when the data scrapping finished (Jan 3, 2024) was 6.605. This default score represents a very coarse qualification for an incoming anime, given that nobody watched it or its fandom is statistically insignificant. Its weight is defined as follows:

(4) c=(mv+m).

When more people scores the anime, C lose relevance in the score. In other words, when s the weight of the scores given by the users grows, c tends to 0.

Finally, the weighted averaged score of an anime is defined as follows:

(5) W=(vv+m)S+(mv+m)C.

This shows the high quality of MAL metrics for popularity measuring. By considering that MAL gather members all around the world and is the most visited anime portal in all internet, any output generated by a method trained with this dataset should be interpreted as popularity across (a huge part of) the internet. However, since this score does not consider the demographic statistics of the voters, it may be not a suitable measure for a profile-based recommender system.

Correlation with extraneous variables

In order to ensure data integrity, it is important to explore whether extraneous variables can influence machine learning predictions. An extraneous variable is a one that is known to have no relationship with the target variable. For instance, in Fig. 1 the synopsis wordcount and the mean character frequency are such variables. The same figure also suggests the possibility of a possible correlation with the target variable (score), which may compromises the integrity of the experiments.

To properly measure how much an extraneous variable can influence the score, correlation coefficients such as Pearson and Spearman can be used, with values ranging from 0 to 1. According to Cohen (1992), it is possible to interpretate the correlation values with the follow rule of thumb: 0−0.29 indicates a small correlation, 0.30−0.49 indicates a moderate correlation and 0.50−1 indicates a strong correlation. Fine-graining, in Abdalla, Vishnubhotla & Mohammad (2021), the authors used another rule of thumb for the Spearman correlation coefficient: 0−0.19: very weak; 0.2−0.39: weak; 0.4−0.59: moderate; 0.6−0.79: strong; 0.8−1: very strong. In this work, the follow hybrid rule of thumb is considered, linking it with the interpretation staten in the Introduction: 0−0.19 for a very weak correlation. Along with a good fit, it means that the used features has no impact on the phenomenon.

0.2−0.29 for a weak correlation. Along with a good fit, it means that the impact of the used features is vague.

0.3−0.49 for a moderate correlation. Along with a good fit, it means that the impact is moderate and more features are required (in this case, features accessible after further investment).

0.5−1 for a strong correlation. Along with a good fit, it means that these features are enough to explain popularity.

Pearson and Spearman will be also used to measure correlations between predictions and real scores (and most detailed explanation is presented further, when discussing the final results). Table 3 presents the correlations between target variable and extraneous variables, showing a moderate correlation with synopsis wordcount, which may compromise the performance of any machine learning experiment. In order to solve this issue, an ablation over the dataset must be performed, in order to reduce this correlations.

Table 3 Pearson and Spearman correlations between the target variable (MAL weighted average score) and both extraneous variables main character frequencies and synopsis wordcounts, across the dataset, before ablating it.

Synopsis wordcount presents a moderate correlation, which may bias the results of any machine learning model.

Variable	Pearson	Spearman	
Main character frequency	0.111	0.145	
Synopsis wordcount	0.332	0.363	

Ablation

Given the high complexity of the data and the task, it is crucial to minimize the reduction of the dataset to avoid further issues in the experiments, such as overfitting. To achieve this, it is important to limit the exclusion of cases to the greatest extent possible. Figure 1 shows an increase in the wordcount growth slope starting from scores higher than 5, indicating that the highest variance in synopsis wordcounts is due to samples with scores of 5 or higher. Moreover, it is clear that higher wordcounts will further increase this variance, leading to the conclusion that ablation should focus considering only the most extreme cases that meet these conditions. Concretely, this is the followed procedure: All animes with scores 5 or higher are elegible to be discarded.

From the elected, only animes with a wordcount higher than a given threshold (after various iterations, 180 words performed better) stayed as disposable. Sort them by wordcount.

Finally, remove the first N of them.

After several trials, N=646 was found to be the minimum number of cases at which the Pearson and Spearman correlations became low, as desired (depicted in Table 4). Table 5 presents the new statistics for the ablated corpus, as shown in Table 1 for the data before ablation.

Table 4 Pearson and Spearman correlations between the target variable (MAL weighted average score) and both extraneous variables main character frequencies and synopsis wordcounts, across the dataset, after ablating it.

Correlations with synopsis wordcounts are now weak (while moderate before the ablation) and, although correlations with main character frequencies grown a little with the ablation, are still far from becoming moderate and hence significant.

Variable	Pearson	Spearman	
Main character frequency	0.190	0.233	
Synopsis wordcount	0.263	0.295	

Table 5 Statistics for the corpus of the synopsis, after the ablation.

Wordcount refers to synopsis. The column Samples still hints a normal distribution.

Score	Samples	Max. Words	Min. Words	Avg. Words	
1–2	2	68	55	61.50	
2–3	6	228	25	112.50	
3–4	11	166	25	80.72	
4–5	77	244	26	78.88	
5–6	911	329	25	87.1	
6–7	3,071	397	25	97.35	
7–8	2,577	277	25	126.18	
8–9	475	233	29	122.02	
9–10	8	174	133	149	
Total	7,138	397	25	101.7	

Evaluated methods

In this section, all the employed neural networks are described. First, the traditional methods equivalents to the proposals from Armenta-Segura & Sidorov (2023), Armenta-Segura (2023), and then the three-input deep neural network with transformers and residual networks.

Traditional model baseline

In order to define a benchmark to evaluate any proposed model, traditional machine learning methods are employed. With this purpose, all texts were vectorized through the Term Frequency-Inverse Document Frequency (TF-IDF) measure with the Scikit-Learn implementation (Pedregosa et al., 2011), while images were tensorized through the PILToTensor method from the Python Imaging Library (Python Imaging Library (Pillow’s Fork), 2022). The result outputs were truncated to size 750 and then concatenated into a 2,250-dimentional tensor which was then feeded to a simple MLP described in Table 6.

Table 6 Specifications of the MLP for the traditional methods.

The last column shows the trainable parameters for each layer: learnable weights (W) and biases (B).

Name	Type	In. Shape	Out. Shape	Act. Fn.	Connects with	Train. Params.	
First	Linear	750+750+750	1,000	TanH	Second	2,250,000 (W)	
						1,000 (B)	
Second	Linear	1,000	500	TanH	Third	500,000 (W)	
						500 (B)	
Third	Linear	500	250	TanH	Fourth	125,000 (W)	
						250 (B)	
Fourth	Linear	250	100	TanH	Lastx	25,000 (W)	
						100 (B)	
Last	Linear	100	1	SoftMax	(logits)	100 (W)	
						1 (B)	

Deep neural network

As anticipated in the introduction, a three-input deep neural network was designed to solve the regression task of predicting the MAL weighted average score given the synopsis, main character portrait and description. In Table 7 an example of the full input of an anime is depicted. For each input, a sequential set of layers was employed: Synopsis: The GPT-2 pretrained neural model on its 124M-parameter modality. Its final layer has as shape 768.

Main character descriptions: The GPT-2 pretrained neural model. It is the same as with Synopsis.

Main character portraits: The ResNET-50 pretrained neural model, flattened at the end. Its last layer is a convolutional layer with an output shape of (7, 7, 2048), where the final 2,048 dimensions corresponds to features exctracted by the model. Due to memory constraints, this work considered two modalities: featureless (only considering the first feature, obtaining a (7,7,1)-shaped tensor who was then squeezed into a (7,7) tensor) and feature-aware (considering all of them). In the first case, the flattened tensor returned a shape of 49, while in the second case returned a tensor with a shape of 100,352. The specific use of each modality in the experiments is precisely determined later.

Table 7 Example of a corresponding sample on the dataset for the anime Witch Hunter Robin, with ID 7 and Score 7.25.

Synopsis (with wordcount)	
Robin Sena is a powerful craft user drafted into the STNJ —a group of specialized hunters that	
fight deadly beings known as Witches. Though her fire power is great, she’s got a lot to learn	
about her powers and working with her cool and aloof partner, Amon. But the truth about the	
Witches and herself will leave Robin on an entirely new path that she never expected! (66 words)	
Main characters	
Name (with MAL ID)	Description (with wordcount)	
Robin Sena (299)	Robin Sena is a soft-spoken 15-year-old girl with … (144 words)	
Amon (300)	Amon is a Hunter and is also Robin’s partner… (412 words)	
Michael Lee (301)	Michael is a hacker and the technical support … (133 words)	
Haruto Sakaki (302)	Haruto Sakaki is an 18-year-old Hunter working with the … (167 words)	
Miho Karasuma (303)	The second in command. Miho is a 19-year-old hunter … (73 words)	
Yurika Doujima (304)	Doujima is portrayed as as carefree, lazy, vain, and … (205 words)	
Total wordcount: 1,134	

Once embedded with their corresponding sequential layers, the main character inputs were concatenated and passed to a multilayer perceptron (MLP) whose specifications are depicted in Table 8 and Fig. 2. The design of this MLP aims to concatenate both main character inputs and to process them into a unified 768-dimentional embedding.

Table 8 Specifications of the MLP for main characters embeddings.

The last column shows the trainable parameters for each layer: learnable weights (W) and biases (B). The layers First and Second have two versions, depending of the ResNET-50 modality, featurless (ftl) or feature-aware (f-a).

Name	Type	In. Shape	Out. Shape	Act. Fn.	Connects with	Train. Params.	
First	Dropout (p=0.1)	768+49	768+49	–	Second	–	
(ftl)							
First	Dropout (p=0.1)	768 + 100,352	768 + 100,352	–	Second	–	
(f-a)							
Second	Linear	768+49	768	TanH	Third	627,456 (W)	
(ftl)						768 (B)	
Second	Linear	768 + 100,352	768	TanH	Third	77,660,160 (W)	
(f-a)						768 (B)	
Third	Dropout (p=0.1)	768	768	–	Fourth	–	
Fourth	Linear	768	768	TanH	(Output layer)	589,824 (W)	
						768 (B)	

Figure 2 Full three-input deep neural network with output shapes.

Input shape of GPT2 for synopsis and main character descriptions was 128 and 256, respectively, and the input shape of the ResNet-50 model was (3,224,224) (RGB channels, and resizing into a 224×224 square). Additionally, refer to Table 8 for details about the MLP for characters, trainable parameters and their activation functions, and Table 9 for details about the Big MLP for classification, trainable parameters and their activation functions. Image credit: https://shmector.com/free-vector/anime/anime_girl/5-0-54 (Shmector.com; CC0 1.0 Universal Public Domain).

Table 9 Specifications of the MLP for classification.

The last column shows the trainable parameters for each layer: learnable weights (W) and biases (B).

Name	Type	In. Shape	Out. Shape	Act. Fn.	Connects with	Train. Params.	
Dropout	Dropout (p=0.1)	768+768	768+768	–	First	–	
First	Linear	768+768	768	TanH	Second	1,179,648 (W)	
						768 (B)	
Second	Linear	768	384	TanH	Third	294,912 (W)	
						384 (B)	
Third	Linear	384	192	TanH	Fourth	73,728 (W)	
						192 (B)	
Fourth	Linear	192	96	ReLU	Fifth	18,432 (W)	
						96 (B)	
Fifth	Linear	96	48	ReLU	Sixth	4,608 (W)	
						48 (B)	
Sixth	Linear	48	24	ReLU	Seventh	1,152 (W)	
						24 (B)	
Seventh	Linear	24	12	ReLU	Eighth	288 (W)	
						12 (B)	
Eighth	Linear	12	6	ReLU	Last	72 (W)	
						6 (B)	
Last	Linear	6	1	ReLU	(logits)	6 (W)	
						1 (B)	

Finally, the main character output is concatenated with the synopsis GPT-2 embeddings and passed through a larger MLP (specifications in Table 9), designed to gradually reduce the dimention to convert it into a singleton suitable for regression over the 1-dimentional MAL score.

Proposed method based on large vision and text models

In order to perform experiments, the dataset was train-test splitted in a 81:100 proportion. The reason behind that specific ratio is that several animes can share main characters. For instance, in Table 10, seven out of the fifty four animes in which the main character Monkey D. Luffy (ID 40) appears are depicted, along with the frequencies of four of his nakamas (companions): Roronoa Zoro (ID 62), Nami (ID 723), Usopp (ID 724) and Vinsmoke Sanji (ID 305). As a consequence, it is important to ensure that all animes with shared main character belongs to the same set so all test samples will corresponds to totally unseen data.

Table 10 As an example of several main characters who appears across several animes (as main characters), this table shows seven animes (out of 54) in which at least one crewmate of Monkey D. Luffy (joined during the East Blue saga) appears.

Anime (with ID)	Luffy	Zoro	Nami	Usopp	Sanji	
One piece (21)	X	X	X	X	X	
One piece film: red (50,410)	X	X	X	X	X	
One piece movie 14: Stampede (38,234)	X	X	X	X	X	
One piece: Taose! Kaizoku Ganzack (466)	X	X	X	–	–	
One Piece 3D2Y:… (25,161)	X	–	–	–	–	
One piece: romance dawn Story (5,252)	X	–	–	–	–	
One piece: cry heart (22,661)	X	–	–	–	–	

Train-test splitting

In order to make the custom split, all animes were grouped into clusters with respect of their shared main characters: if two animes shared a character, they were collocated in the same cluster. It is worth to note that, for each cluster, it is possible to find two animes with no shared characters, but a chain of animes who shares characters and who connects them. For that reason, the next recursive algorithm was employed to generate the clusters, obtaining 3,987 (out of 7,138 samples): For each anime, get all the other animes who shares a main character. Assign to all of them a number (cluster name).

Make a second pass across all the dataset. This time, if two animes shares a cluster name, assign a new cluster name to them.

Repeat this algorithm until all animes have associated a single cluster name.

Finally, the train-test splitting was performed randomly, but each cluster was completely contained withing a single split. The training set has 5,852 samples ( 81.5%) while the test set has 1,286 samples (18.5%). Fortunately, despite the random splitting, both sets obtained very similar statistics, as evidenced in Table 11 and Fig. 3 for the synopsis wordcount, and Table 12 and Fig. 4 for the character wordcount.

Table 11 Statistics for the training and test set of the synopsis.

As in Table 5, once again the Samples column hints at normal distributions in both splits.

Score	Samples	Max. Words	Min. Words	Avg. Words	
Training set	
1–2	1(0.02%)	55	55	55	
2–3	5(0.09%)	229	25	125	
3–4	9(0.15%)	166	25	82.4	
4–5	63(1.07%)	244	26	76.52	
5–6	743(12.7%)	329	25	81.42	
6–7	2,519(43.05%)	384	25	97.7	
7–8	2,128(36.36%)	100	29	137.77	
8–9	378(6.46%)	180	29	107.08	
9–10	6(0.1%)	174	134	149	
Total	5,852(100%)	581	25	101.37	
Test set	
1–2	1(0.08%)	68	68	68	
2–3	1(0.08%)	52	52	52	
3–4	2(0.16%)	87	61	74	
4–5	14(1.09%)	202	27	88	
5–6	168(13.06%)	254	26	92.78	
6–7	552(42.92%)	397	25	97.01	
7–8	449(34.91%)	277	25	114.58	
8–9	97(7.54%)	233	35	136.98	
9–10	2(0.16%)	165	133	149	
Total	1,286(100%)	397	25	96.91	

Figure 3 Mean synopsis wordcount in the training (left) and test (right) set.

X-axis represents the floor-rounded MAL score. Y-axis is the mean word counts on synopsis per score.

Table 12 Statistics for the characters.

Split	Total characters	Max. Words	Min. Words	Avg. Words	
Train set	14,108	3,498	5	110.78	
Test set	3,188	2,286	5	114.69	
Total	14,602	3,498	5	112.54	

Figure 4 Mean character frequency in the training (left) and test (right) set.

X-axis represents the floor-rounded MAL score. Y-axis is the mean amount of characters per score.

Experimental setup

All neural networks were implemented with PyTorch 1.10.1 (Paszke et al., 2019) and run on an NVIDIA Quadro RTX 6000 GPU with 46 GB of VRAM. In the case of the three-input deep neural network, the neural network used the Huggingface models (Huggingface, 2022a, 2022b) and all inputs were processed with the Huggingface tokenizers and image processors. In the case of text, the GPT2Tokenizer returned tensors with shapes the max lengths depicted on Table 13 ( 128 for the synopsis and 256 for the character descriptions). In the case of images, the AutoImageProcessor method returned tensors with shapes (3,224,224), corresponding to an image resize into 224×224 and the three RGB color channels.

Table 13 All employed parameters in the experiments.

Parameter	Value	
Seed	42.	
Synopsis pretrained model	‘GPT2’.	
Char. Desc. pretrained model	‘GPT2’.	
Image pretrained model	‘microsoft/resnet-50’.	
Synopsis Tokenizer Max. length	128 (GPT2tokenizer).	
Char. Desc. Tokenizer Max. length	256 (GPT2tokenizer).	
Image processors parameters	Default (AutoImageProcessor).	
Image processor resize shape	224×224 px	
Image processor color channels	RGB	
Optimizer	Adam with weight decay (AdamW).	
Opt. learning rate	5e−2.	
Opt. Epsilon param	1e−8.	
Loss function	Mean squared error (MSEloss).	
Batch size	16.	
Epochs	5 ( 30 for traditional vectorizations).	

Five experiments were conducted: the benchmark with the traditional vectorizations (Trad) aiming to replicate an experiment analogous to the prior work (Armenta-Segura & Sidorov, 2023; Armenta-Segura, 2023) and the three-input deep neural network with all inputs (Full), with only synopsis (GPT-2 + MLP), only portraits (ResNET-50 + MLP), and only descriptions (GPT-2 + MLP). For all inputs, the featureless modalitiy of the ResNET-50 model is employed due memory constraints as mentioned previously. For synopsis and descriptions (text inputs), the large MLP depicted in Table 9 was modified by removing the First layer. For portraits (image inputs), the small MLP depicted in Table 8 served as the base, using the feature-aware modality of the ResNET-50 model and with two additional linear layers with ReLU activation functions and output shapes of 384 and 1, respectively, added for regression purposes. Finally, for the traditional methods, the MLP depicted in Table 6 was used. All experiments utilized the same hyperparameters depicted in Table 13, except for the epochs, since the experiment Trad employed 30 instead of five as the other experiments. Additionally, the scores were scaled to range between 0 and 1, with 0 assigned to the minimum score and 1 to the maximum.

Results

Results are depicted in Table 14. Learning curves for each experiment can be found in Fig. 5. As anticipated in the Introduction, the superior experiment was “Full”, with the best metrics, the best fitting on the learning curve and without either overfitting nor underfitting after the first epoch. All non-traditional experiments outperformed the MSE benchmark.

Table 14 All experiments, sorted by mean squared error (MSE): the lower the value, the best the result.

Moderate correlations are highligted with bold.

Popularity prediction	
Experiment	Input	MSE	Pearson	Spearman	R2	
Full	All	0.012	0.382	0.362	0.142	
GPT-2+MLP	Syn.	0.012	0.306	0.327	0.083	
GPT-2+MLP	Char. (Desc.)	0.012	0.304	0.306	0.094	
ResNET-50+MLP	Char. (Img.)	0.013	0.293	0.295	0.013	
Trad (benchmark)	All	0.415	0.083	0.105	−37.591	

Figure 5 Learning curves for all experiments.

From top-left to bottom-right: All inputs (good fit), Syn+MLP (slightly underfit), Char+MLP (slightly underfit), Img+MLP (notable underfit) and traditional methods (catastrophic underfit, as expected due the size and complexity of the dataset).

Discussion

Three main aspects must to be evaluated in order to interpretate the results: accuracy, correlations and fitting of learning curves.

Accuracy

Recall that the MSE is calculated with the square values of the differences between the real and the predicted scores. Hence, a value closer to 0 might seem as the best scenario possible, as long as the learning curve does not evidence bad fit.

The traditional benchmark yielded a MSE of 0.415, which is more than 34 times the best result ( 0.012) and more than 31 times the worst result by neural models ( 0.013, only images) as showed in Table 14. Hence, it is possible to state that the accuracy of the proposed methods is good (when compared with the other experiments).

Correlations and R2

The same rule of thumb presented during the data exploration is employed here. However, in order to understand its true implications with the results, the follow subsections dives further into both Pearson and Spearman coefficients.

Pearson

According to Berman (2016), the Pearson correlation coefficient aims to measure linear correlation: the likelihood of the existence of a linear function who maps the independent variable (predicted scores) into the observed variable (MAL scores). If the correlation is near to 1, it exists such linear map. When the learning curves have a good fit, this linear relationship also illustrates how the employed input can impact the popularity of an anime.

This is also a most flexible metric to evaluate performance than MSE since considers any linear correlation, beyond the MSE’s considered identity mapping. For instance, if a model predicts any score X as 2X, its MSE will be bad but its Pearson coefficient would be 1 (the best possible) and the model could solve the task by considering the inverse mapping (or an approximation, if not invertible). In general, if T is a linear map and, for each score X, the model predicts T(X)−δX for a very small δX (i.e., δX lies in a tiny neighborhood of ratious an error ε), then the Pearson coefficient will be near to 1−ε, suggesting that the accuracy of the model can be enhanced the most by adding linear operations to its output, aiming to simulate T−1+mean(δX). However, if the error ε is big, the probability entropy of T−1+mean(δX) might be higher and hence T may be not unique, meaning this a lack of linear correlation and therefore a lack of information from the neural network (and hence the input features) to solve the task, whenever the learning curves presents a good fit.

In this case, only images achieved a very low Pearson correlation of 0.293, meaning that performance cannot be enhanced with solely MLPs although empirical evidence suggests that leveraging larger versions of ResNET must cushion this issue. Moderate but weak correlations are achieved from all other experiments, suggesting the existence of some small parts when a linear map as described before can be approximated.

Spearman

According to Statstutor (2025), the Spearman correlation coefficient is similar to Pearson’s, but adds monotonicity and works well whenever the data distribution is similar to normal. In Fig. 6 and Tables 5 and 11 are shown how this condition holds for this data. It is calculated by the same formula as Pearson, but with the following steps: Order all independent variables from lower to higher.

Order all observed variables from lower to higher.

Match them with this new order.

Figure 6 Number of animes per MAL score (rounded).

From left-right, top-bottom, train split, test split and both. Each split of the dataset shows a normal-like distribution, which motivates the use of Spearman over other rank-based correlation coefficients, such as Kendall’s Tau.

This is made through ranks: the lowest variable receives the rank 1, the 2nd lowest the rank 2 and so on, until the highest variable, who receives the last rank. Finally, two variables are matched if they share the same rank. If predictions are not monotonic with respect of the observed variables then this procedure will remove the original matching. Spearman correlation is, hence, Pearson correlation but calculated with this monotonic matching. In this case, Spearman coefficients are very close to its Pearson homologous, meaning that any slightly existent linear correlation is also monotonic.

R2 score

The R2 score measures the proportion between the variance of the error and the variance of the observed variables. If its closer to 1 then the predictions are totally correlated with the golden standard. If it is negative, the model fails totally on capture any kind of correlation and performs worst than a horizontal line (i.e., making always the same prediction). Values from 0 to 0.5 can be considered mediocres, 0.5 to 0.75 are acceptable and 0.75 to 1 are good. However, it is important to also consider how the learning curves fitted during the training of the model in order to properly evaluate the performance.

In this case, all R2 measures from all experiments (excluding the Traditional Benchmark) were between 0.01 and 0.15, with no negative performances. In order to properly determine the consequences of such scores, the next section will study the learning curves.

Fitting of learning curves

As evidenced in Fig. 5, all training curves for all cases are not underfitting and have an acceptable decrease rate. However, the validation curve only presents a good fit in the case of all inputs, while shows a slight overfit when considering only a subset of them. The traditional benchmark, on the other hand, presents a catastrophic overfitting due the failure of this kind of methods on learn something from the data, which brings a solid proof of the hypothesis from Armenta-Segura & Sidorov (2023). This also coincides with their R2 score of −37.591.

Image inputs

From all the particular cases, this input showed the widest gap between learning curves. Along with holding the lowest R2 score and correlation coefficients, the leverage of larger models is necessary to categorically determine how much impact and scope can have this kind of data alone for predicting popularity of an anime.

Text-based inputs

Both text inputs showed a similar small gap. This, along with their similar metrics of weak-moderate correlation, implies that there is still a small room for improve the models. However, the gap is very small, hence theorizing that the existence of a weak synergy between plot summaries and main character descriptions with features available after further investment it is reasonable and could be very close to reality.

All inputs

All inputs yields a good fit starting on Epoch 2. Hence, the room of improvement for this model may be insignificant and hence it is valid to state that: All features together are relevant for determining the popularity of an incoming anime.

However, they are not decisive enough for properly define and explain it.

As a result, there must exists synergy between all these features combined and others that become available after further investment, as anticipated in the Introduction. In conclusion, although there might be significant room for improvement in the image embeddings, and limited room for improvement in the text embeddings, when considered together, there is just an insignificant gap to enhance the multimodal model, making the insights from this study sufficiently reliable.

A memory constraint with transformer-based models

The proposed GPT-2 model presents the third most pronounced overfitting, only behind the main character portraits and the Traditional Benchmark. Table 13 shows that the maximum length for the tokenizer of main character descriptions is 256. However, some of the anime synopsis, when concatenated, can surpass 1,000 words, as shown in the example in Table 7. Consequently, the small gap between learning curves in the text-based inputs can be explained by this loss of crucial information. Hence, further experiments must to be done in order to answer whether if the employment of larger LLMs such as Llama2 (Touvron et al., 2023) or GPT4 (Achiam et al., 2023) vs. the use of tricks to avoid this situation (such as processing the text on parts) are the way to eliminate this gap.

Conclusions

In this article, one of the most robust free datasets for anime popularity prediction only with features available before huge investments was introduced, relying solely on freely available internet data. To explore this dataset and its scopes, a deep neural network leveraging GPT-2 and ResNET-50 was proposed. To measure the performance of the models, MSE and R2 Score were employed. The best MSE achieved was 0.012, significantly lower than the benchmark MSE of 0.415, and the best R2 score was 0.142, which far outperformed the benchmark R2 of −37.591.

The main goal of this work was to answer the follow research question:

RQ: What are the scopes and impact of the available features before making huge investments, in relation of the popularity of an anime?

Pearson and Spearman correlation coefficients between the predicted and the real scores were calculated in order to answer it: low correlations, along with a good fit of the learning curves, means no scope and an absence of impact between the employed features (embedded via an MLP into a single dimension, equivalent to a popularity score) and popularity. Moderate correlations with a good fit suggests the existence of a synergy between this features and other available after further investments (for instance, features similar to those presented in Wang et al., 2019; Airen & Agrawal, 2023). Finally, high correlations with a good fit means that these features are capital and decisive with respect of popularity, and hence it is possible to design truthful models for solving the task relying solely on this family of features. In this work, the experiment considering all the features obtained a good fit, but moderate correlations, demonstrating that these set of features are not enough to explain the popularity of an anime. This encourages the follow further work: Studies of synergy for developing evaluation benchmarks and standards for production: This work demonstrated that features accessible before huge investments are relevant when determining popularity of an incoming anime. However, it also demonstrated that they are not decisive. Its outcomes also suggested the existence of a relevant synergy with other kind of features, which leads to the further work of determining how this synergy works. On the spirit of aiding companies and animation houses to avoid financial failures, unveiling the mysteries behind this synergies can also leads on the development of benchmarks and standards who may ensure production quality, and hence maximize the probability of cultural and financial success.

Explaining the relevance of this features: This work showed, via correlation coefficients, the relevance of features before huge investments for popularity. Explainable AI (XAI) techniques could be leveraged over this model in order to explain how and at which levels each feature is relevant. This could answer several interrogants about how to guide the production of an anime, given an initial idea which may add to the idea from the previous point.

The memory problem with main character descriptions: Truncating main character descriptions to 256 tokens significantly impacts the results due the information lost, reflected in the gap between learning curves. This problem, closely related with the memory constraint of transformer-based models, has these two paths to overcome it: – To experiment with other language models with less memory requirements. E.g., selective state spaces such as Mamba.

– To adjust the input to embed each main character individually. These individual tensors can then be processed in a miriad of ways, such as averaging them. The information lost after such processes is an important topic to be explored.

Due to the memory limitations of transformer-based models, it is not trivial to increase the tokenizer size. Therefore, further work should explore more memory-efficient methods capable of capturing all information with similar learning capabilities (e.g., Mamba; Gu & Dao, 2023) or find other alternatives to include all main character information simultaneously (e.g., generating mean tensors and assessing how much relevant information is truly lost from doing that). The research question to answer here is what are the reasons behind the gap presented between the training curves of GPT-2 embeddings, which also connects with the next further work. Larger models for processing the inputs: GPT-2 can be upgraded to GPT-4 or Llama3, and ResNET-50 can be enhanced to its larger brother ResNET-152, a very deep pretrained CNN or a vision transformer such as Image-GPT. The moderate Pearson and Spearman coefficients along with the gap between learning curves indicates that leveraging larger models can benefit the results. Although it seems to be impossible to overcome a moderate correlation, larger models could lead to a most exhaustive processing of the features and, with the help of XAI, provide soundest explanations about the influence of these features into popularity.

Supplemental Information

Supplemental Information 1 Codebook.

ChatGPT 3.5 was employed to fix the style and grammar issues in a few paragraphs. The suggestions were reviewed and revised again as necessary.

Additional Information and Declarations

Competing Interests

The authors declare that they have no competing interests.

Author Contributions

Jesús Armenta-Segura conceived and designed the experiments, performed the experiments, analyzed the data, performed the computation work, prepared figures and/or tables, authored or reviewed drafts of the article, made the submission, and approved the final draft.

Grigori Sidorov performed the experiments, analyzed the data, prepared figures and/or tables, authored or reviewed drafts of the article, suggested the journal and the reviewers, and approved the final draft.

Data Availability

The following information was supplied regarding data availability:

The data is available at Zenodo: Armenta-Segura, J. (2024). Popularity Prediction in Anime with Deep Learning (with dataset) [Data set]. Zenodo. https://doi.org/10.5281/zenodo.13835115.

The detailed instructions for reproducibility are available at GitHub and Zenodo:

- https://github.com/JesusASmx/Popularity-Prediction-in-Anime-with-Deep-Learning.

- Armenta-Segura, J. (2024). Popularity Prediction in Anime with Deep Learning. Zenodo. https://doi.org/10.5281/zenodo.13824604.

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
