# Peer review of "Anime popularity prediction before huge investments: a multimodal approach using deep learning"

_PeerJ Computer Science, doi:10.7717/peerj-cs.2715_

## Round 0.1 · original submission · Major Revisions

I am sure that the authors will be able to address all critical comments of the reviewers.

Reviewer 1 ·

Basic reporting

The authors have presented a deep-learning approach for predicting the popularity of Japanese anime. Overall, the English of the paper is clear and unambiguous. The introduction and background sections are well-referenced and relevant.

Experimental design

Experiment details are sufficient and can easily be replicated. The data preprocessing details are well-discussed. All evaluation metrics and the proposed model are described adequately.

However, few suggestions to further elaborate:
In Tables 4, 5 and 6, the authors should also include the column of trainable parameters.

The authors should elaborate on the architecture of GPT-2 in Fig. 5. Also, mention the linear layer shape etc. Furthermore, correct the spelling mistake of "linear".

Validity of the findings

Are the experiments and evaluations performed satisfactorily? Yes

Does the Conclusion identify unresolved questions/limitations/ future directions? Yes

Additional comments

Some comments are:
1. In the abstract, the authors wrote "On the other hand, the methods consists on a deep neural network architecture who leverages GPT-2 and ResNet-50 to embed the data and process it through a multilayer perception"

This sentence needs revision for better readability. A suggestion is "The framework consists of a deep neural network that leverages GPT-2 and ResNet-50 to embed the data and process it through a multi-layer perception."

Also, change in the introduction section (lines 61-62)

2. Change the term PILtotensor to PILToTensor throughout paper

3. Some tables have captions at the bottom. Put captions of tables on top, while the figures are at the bottom.

Reviewer 2 ·

Basic reporting

1. The authors describe the topic and its importance adequately. However, when discussing the related work they missed important work which also addressed similar topics such as [1].
[1] S. M. AlSulaim and A. M. Qamar, “Prediction of Anime Series’ Success using Sentiment Analysis and Deep Learning,” in 2021 International Conference of Women in Data Science at Taif University (WiDSTaif ), Mar. 2021, pp. 1–6. doi: 10.1109/WiDSTaif52235.2021.9430244.

Experimental design

1. The traditional method based on previous work (17, 9 as cited in the text) is cited as such in the Introduction but should also be explicitly cited and described as previous work in the Experimental Setup.
2. In lines 167 to 170 the authors describe the neural networks used to extract features from the image and text inputs but it is not clear how do they obtain the outputs with shape 768 and 7x7 for GPT-2 and ResNet-50 model.
3. Also, usually images are preprocessed (resized to a same size) in order to be processed as a batch, but we didn’t see it mentioned in the text.
4. It is not clear what is the correlation between the inputs and correlation coefficients supposed to measure. For example, the Pearson correlation coefficient between an independent variable such as the number of users giving a score and the dependent variable for the MAL score is easy to understand, but in this case, there is not a single value to use as the independent variable x to correlate against the dependent variable y corresponding to the MAL scores. Explaining what is the independent variable and how was it computed is important to understand the significance of these correlation coefficients.

Validity of the findings

1. Our main concern with the findings is how it seems to be a correlation between the mean word count (for the synopsis and character descriptions) and the score, as shown in Figure 3 and 4. These two plots seem to indicate that in average animes with higher scores have longer descriptions, while ones with lower scores have shorter descriptions. If that’s the case a machine learning model could easily predict scores based on length of reviews without counting into account the descriptions themselves. We suggest the authors to consider an ablation where they control the text lengths by only using animes with a limited range of mean word count to validate the predicted scores are due to the synopsis themselves and not due to external factors such as the length of the text.
2. Also, the authors describe underfitting in most of the sub-figures of Figure 6, but based on our observations it seems most of them show overfitting since there is a gap between the training and validation loss, and only the last one shows both underfitting and overfitting. Therefore the discussion regarding using larger models to improve results is not intuitive.

Additional comments

Strengths:
1. The topic is interesting and the contribution of a dataset to measure the popularity of an anime could be useful for future work.

Others:
1. Presentation and descriptions could be improved.
1.1. Table 8 represent statistics for the synopsis but the caption is only “statistics for the training and test set”. It should be “Statistics for the training and test set of the synopsis”. In general, Tables and Figures should be able to stand on their own through their captions and descriptions so that the readers clearly understand what is being conveyed in them.
1.2. In Figure 3 and 4 the Y axis should represent the same quantity (number of words “word count”) but they are labeled differently.
1.3. In Figure 5 the text in the BigMLP is not aligned with the “boxes” and the GPT-2 module is represented as a sequence of boxes with arrows pointing in different directions. This should be modified to either describe it as a blackbox or employ a figure similar to their respective paper.
1.4. In Figure 6 the authors could include the “input” at the top of each figure so that readers could clearly discern what does each figure represent instead of having to look at the legend.
1.5. Since there’s different variants of GPT-2 the authors should specify the one they use (usually described in terms of parameters).

2. The background section could be renamed as methodology.

---

## Round 0.2 · Major Revisions

Unfortunately, the introduced changes did not satisfy the reviewer. I propose you to continue editing the manuscript in light of the proposed changes.

Reviewer 2 ·

Basic reporting

1. We are not the authors behind the mentioned work [1] but at least including this in the paper and how is it different or filling a gap compared to this previous work is critical for completion.
[1] S. M. AlSulaim and A. M. Qamar, “Prediction of Anime Series’ Success using Sentiment Analysis and Deep Learning,” in 2021 International Conference of Women in Data Science at Taif University (WiDSTaif ), Mar. 2021, pp. 1–6. doi: 10.1109/WiDSTaif52235.2021.9430244.
2. Line 160 (in the file with tracked changes), it should be R squared or coefficient of determination not root squared.
3. In Line 215 change “Neural Networks” section to “Evaluated Methods” to convey this section includes both the traditional baseline
4. Line 219 change “Traditional models” to “Traditional Model Baseline”.
5. Line 226: Deep Neural Network section describing the proposed method incorporating GPT-2 and ResNet50 should be renamed to “Proposed Method Based on Large Vision and Text Models” or something similar to easily point to the reader that this section describes the proposed system.

Experimental design

1. In the previous review we mentioned it is not clear how they obtained the output features for the GPT and Resnet model. They partially explained these but for the Resnet’s output it should be a 3D tensor with 768 channels and 7x7 2D structure with an overall shape 768x7x7 for ResNet50 instead of only 7x7 so we would like authors to explain how is this 3D tensor is converted into only 7x7 features.

Validity of the findings

Last time we mentioned the following "Our main concern with the findings is how it seems to be a correlation between the mean word count (for the synopsis and character descriptions) and the score, as shown in Figure 3 and 4. These two plots seem to indicate that in average animes with higher scores have longer descriptions, while ones with lower scores have shorter descriptions. If that’s the case a machine learning model could easily predict scores based on length of reviews without counting into account the descriptions themselves. We suggest the authors to consider an ablation where they control the text lengths by only using animes with a limited range of mean word count to validate the predicted scores are due to the synopsis themselves and not due to external factors such as the length of the text".

The authors' response as follows: "We conducted the ablation, and the results did not change significantly (except for the expected worsening of the learning curve fit). Upon further investigation, we realized that the issue arose due to the use of Kendall’s Tau as the correlation coefficient. After diving deeper, we identified an oversight on our part: Kendall’s Tau is primarily useful when the data distribution is non-normal. Since our data follows a normal distribution, using Kendall’s Tau could lead to misleading insights (such as the existence of a correlation between wordcounts and score). Therefore, we replaced it with the rootsquare method and explained the linear rank correlation using solely Spearman, who is most suitable for scenarios where the data distribution is normal. The outcomes of this discussion, along with the normal distribution of the data, are in the caption of Figure 6 (new enumeration)." did not address our concerns.

They mentioned how the outcomes of this discussion and the normal distribution of the data are in the caption of Figure 6 but Figure 6's caption does not include a discussion on how "truncating" the text lengths by only using animes with limited range of word count to validate that the predicted scores are due to the content in the synopsis themselves or the character descriptions and not due to correlation between the length of the description itself and the scores themselves (as currently shown in Figure 4 and 5 according to the new numbering).

---

## Round 0.3 · Minor Revisions

I am sure that the authors will easily finalize the manuscript to introduce the last improvements.

Reviewer 2 ·

Basic reporting

The authors have addressed missing relevant work.

Experimental design

The authors have addressed the original concerns with the experimental design.

Validity of the findings

The results show the promise behind the proposed method under fair settings.

Additional comments

Outern variables in line 202 and the following paragraphs should be "extraneous variable". Outern is not a standard term in English.
In line 267 the authors describe the output of the ResNet as 7x7 spatial positions with 2048 channels then describe two modalities: feature-aware and featureless, mentioning that featureless "discards them". By discarding it do you mean you apply some kind of reduction (either average or max) across the channel dimension so that the output becomes 7x7? If that's the case consider describing if the reduction is average or max or if the proposed method only uses a single channel (let's say the first one) to clarify that.

---

## Round 0.4 · accepted · Accept

I have assessed the revision myself. I am happy with the current version.
The manuscript is ready for publication.